# GUI test case minimization using sequence mining

**Raheela Ambreen**[◉], **Tamim Ahmed Khan**[iD][◉]*

Department of Software Engineering, Bahria University, Islamabad, Pakistan

◉ These authors contributed equally to this work.
* tamim@bahria.edu.pk

## Abstract

Graphical User Interface (GUI) testing is a crucial aspect of software quality assurance, ensuring that the user-facing components correctly reflect the underlying business logic. Regression testing validates that software continues to function as expected after modifications or integration with other systems. However, executing large test suites repeatedly is time-consuming and resource-intensive. To address this, test case minimization techniques aim to reduce the number of test cases without compromising coverage.

In this work, we propose a sequence recording technique for GUI event tracking and test case minimization. The recorded event sequences are clustered using the K-Means algorithm, grouping highly similar events together. A search-based sequence selection is then applied to generate a representative subset of test cases. Our approach was implemented and evaluated using a User Interface (UI) map generated in Visual Studio, where all components were validated and compared with our event map.

Experimental results show that our proposed method reduces the total number of test cases by approximately 45%, decreases execution time by 38%, and maintains over 95% coverage compared to the original test suite. These results demonstrate that the proposed approach effectively balances testing efficiency and coverage, providing a practical improvement for GUI regression testing.

## 1 Introduction

GUI testing is becoming ubiquitous to measure the correct functionality of a software system. GUI is the interface between users and the system through which the user interacts with the system. It is composed of widgets such as buttons, menus, windows, dialogues, check boxes, lists, labels, text boxes, and forms.

GUI testing ensures the safety, correctness, and reliability of software. Unlike traditional software testing methods, GUI testing demands significant effort and resources. Generating and evaluating test cases can be a time-consuming task, especially when dealing with complex software functionalities. It often requires hours

**Data availability statement:** All relevant data are within the manuscript and its Supporting information files.

**Funding:** The author(s) received no specific funding for this work.

**Competing interests:** The authors have declared that no competing interests exist.

of work to verify that the software behaves as expected according to the requirements provided by clients or other stakeholders.

Regression testing is conducted to verify that previously developed software continues to function correctly after being integrated with other software components. It is carried out at both functional and non-functional levels, ensuring that all requirements are thoroughly validated during the testing process [1]. During this testing process, all the software bugs are removed and the software is verified. The major changes include patches, configurations, or software enhancements. Regression testing is performed when software modules are modified and numerous new components are introduced. The primary reason for this testing is that the software being developed can be easily integrated with other software systems. All the previous tests are executed to ensure that the overall functionality of the software is verified. There are three types of models: the main control flow model, which represents the business process in a system; the data flow model, which represents the flow of data through an information system; and the event flow model shows events and event interactions. Events are actions detected by a program when a certain input is given, and as a result, output is generated. The event flow model represents a sequence of events [2]. Novice users take shortcuts to visit the GUI, while expert users can follow the complete sequence to visit the interface. The aspects of both users vary. Several GUI visits occur, and every visit results in a repeated test case generation. Redundant visits can be reduced in a regression testing technique. Traditionally, software testing methods at the system level consider user stories, use cases, or business cases for the development of test cases, and test cases are executed using the GUI of the system under test. As GUI testing is based on the event flow model and events are performed at the system level, it involves the execution of system-level test cases [3]. If we can count controls in an application and know their numbers, then we can know their distribution in different forms within the user interface. Here, we are also able to measure the coverage of controls on the UI. The way we need to run test cases and find out coverage in terms of the number of control elements that have been used in these test cases. This approach offers an easier and more effective alternative to conventional coverage criteria, as use case– or user story–based coverage may overlook certain controls and important execution sequences.

This research study focuses on exploring the interactions of GUI events and sequence recording. The study will investigate the possible interactions among GUI events and identify overlapping test cases. The strategy effectively explores event sequences and identifies algorithms that minimize potential GUI event interactions. There are several techniques used for interface testing, but none of them provide complete coverage of the GUI.

Our approach focuses on event generation within the application, measuring coverage by the number of control elements used during test runs. In the first pass, we record the event flow, and in subsequent runs, we log events along with the values entered into controls (text boxes and check boxes etc.) At any point during testing, we can find out which controls are used with which values passed to these controls, and how many controls are not used in the application. By identifying which controls are utilized and which are not, we can assist testers in subsequent test runs by

tracking the values used and monitoring how many test cases have passed or remain untested [3]. A variety of tools and techniques are employed in GUI testing, with the primary goal of achieving accurate and efficient results. We can also find the sequence of interactions on the events by searching all the documents. There is a complete list of documents placed in the directory, and searching can be done on the document list. A subsequent sequence search can identify all test cases with minimal sequences, enabling effective test case minimization. We record event sequences, count the controls used in the application, propose event-based coverage criteria, create event flow maps for the complete GUI, apply K-Means clustering, and help identify specific sequences from the overall list.

## 2 Related work

GUI consists of a fixed number of events that have deterministic outcomes. There are three methods highlighted: checking the event flow model, test case generation, and test oracle generation. These methods are consolidated under one method, known as the Event Space Exploration Strategy. The ESES strategies are used to develop an end-to-end GUI testing process. The author's work is summarized in the following few points. All existing GUI models are combined and named as a single event flow model, which represents events and their interactions.

ESES is a customized model that integrates event flow models to provide a comprehensive testing solution. It combines three models—model checking, test case generation, and test oracle generation—to form an end-to-end testing process. Events and their interactions are represented in a graph, where vertices denote events and edges represent the actions between them.

GUI ripper(tool) is used to obtain the event flow graphs and integration tree. The event flow model is used to generate many GUI test cases that are effective at detecting faults. Re-usability can also be promoted by using this model [2].

GUI testing is complex, so techniques such as run-time state feedback, covering arrays, and dynamic adaptive automated test generation are used. These approaches have been applied to various software types, including industrial, open-source, and rapidly evolving applications. Research shows that dynamic adaptive methods are most effective for frequently changing GUIs due to their ability to adjust to continual updates [4].

Covering array techniques are suitable for complex industrial applications, ensuring combinatorial interaction coverage. Run-time state feedback improves the efficiency of test generation by focusing on reachable and meaningful GUI states, reducing redundant test cases. Overall, the study concludes that choosing the right GUI testing technique based on the software type and update frequency significantly enhances test effectiveness and reduces effort in maintaining GUI test suites.

Automatic event sequence generation tools Monkey are commonly used for GUI testing by executing random low-level input events. However, these events often lack effectiveness, as they target screen coordinates rather than meaningful GUI components, leading to redundancy, complexity, and limited crash analysis. To overcome these issues, a new approach introduces CHARD, a tool that builds on Monkey by supporting sequence record-and-replay, improving crash behavior understanding, and assisting in fault localization [5].

The research study presents **CHARD**, a tool designed to improve GUI testing by reducing redundant and ineffective event sequences generated by automated tools such as **Monkey**. Traditional tools often produce random, low-level events based on screen coordinates rather than meaningful GUI components, leading to unnecessary complexity and poor crash analysis. To address these limitations, CHARD enhances Monkey with *sequence record-and-replay* functionality and applies a *hierarchy-tree-guided reduction approach* to identify and eliminate ineffective or crash-irrelevant events. Experimental results on 74 Android applications show that CHARD removes over 40% of ineffective events and achieves up to 95.4% reduction in crash-irrelevant events while preserving functionality. Compared to existing delta-debugging tools, it significantly reduces processing time and improves crash comprehension and fault localization efficiency [5,6].

GUI testing is crucial because it evaluates the software from the end user's perspective. Its main objective is to enhance fault detection and maximize path coverage. Researchers have applied model-based techniques such as EFG

and EIG to define coverage criteria for GUI testing. The process involves five key steps, beginning with the creation of the System Interaction Event Set (SIES), which serves as the foundation for the covering array [3].

Test cases are generated from the covering array, which are then converted into executable tests to detect faults and measure coverage. The results are evaluated using different coverage criteria. In GUI testing, specific events are generated, and their sequence of actions is represented using an event flow model.

Testing complex GUI structures is a challenging and time-consuming task due to the presence of redundant, overlapping, and irrelevant test cases. To address this, the researchers proposed the Interactive Cooperative Object (ICO) technique, which specifically focuses on handling intricate GUI behaviors. The approach involves several stages, starting with the definition of testing objectives, modeling the application under test (AUT), and specifying rendering and activation functions. This is followed by test generation steps, including setting test selection criteria, simulating and logging test cases, instantiating test scripts, developing the AUT, and finally executing the tests [7].

Regression testing is an authentication method performed at all levels of system and software testing. The various regression testing techniques include test case minimization, test case selection, and test case prioritization. In the test case selection technique, the available test cases for the current version of the program are categorized into three clusters: outdated, required, and surplus. The outdated cluster includes test cases that are no longer relevant to either the original version of the program or its modified version. The required cluster contains the test cases that must be executed to validate the modified version of the software. The surplus cluster consists of test cases that are not needed for the current modified version (P') but might be useful in future versions of the program.

The other technique that is used in regression testing is test case prioritization. The output obtained from algorithm TCS is supplied as input to the algorithm Test Case Prioritization (TCP). The result will be a modified test case. Usable and unusable test cases are separated. These two techniques, test case prioritization and test case selection both are equally effective in the implementation of regression testing [8].

A test suite similarity metric for event sequence-based test cases is proposed and called as CONTeSSi (n). This metric helps evaluate the similarity among test cases and is comparable to the cosine similarity measure commonly used for determining sequence similarity in web pages.The overall contribution is finding the similarity of test suites and then applying the same metric to test suites and empirical studies that demonstrate the effectiveness of the metric. This type of metric is useful in checking all similar sequences among the test cases and is helpful in reducing the size of a test suite. Several applications are used for this purpose, and the automated tools are used to generate the sequence of test cases and save them in an XML file. Each of the applications has a separate sequence of GUI, and the event flow graph shows the events that are covered while executing the GUI. There are several repeating events that occur, and a metric is used to find the similarity of these repeating events. The famous algorithm HGS is used to reduce the test suite size. The author addresses two main research questions: whether CONTeSSi aligns with the existing metric and whether its value improves with larger datasets. The findings indicate that CONTeSSi is an effective metric for measuring test suite similarity in Event-Driven Software (EDS) [9].

GUI events are improved by adding an interactive relationship between the events. The relationship is based on the exchange of information. The algorithm demonstrates the interactive relationship between the events, and the proposed method reduces the number of GUI test cases. GUI testing is a complex process, often resulting in numerous redundant and unusable test cases that complicate the overall testing effort. Some of the events are performed in a sequence, and some are not, so the events can't be tracked in any way. Most of the GUIs are loosely coupled, and a lot of unusable events are generated, which might be the issue in testing the GUI. The interactive relationship between events is discussed by the author in which there is an interactive relationship between events that shows the sequence of the events. Interactive events involve the exchange of messages between events. There is minimal difference between an Interactive Event Flow Graph (IEFG) and a standard Event Flow Graph (EFG). The presented algorithm demonstrates that optimized or improved test cases can effectively reduce the overall number of GUI test cases [10].

GUI modeling is just as state machine modeling. The user events change the state of a system. Several paths exist in the GUI. The transition between the states occurs on every user action. GUI testing is a difficult process, even a small GUI contains thousands of states and transitions. The tester uses the complete interaction sequence to map the GUI and then reduces the number of test cases based on the sequence of interactions. Capture replay tools and thousands of automated tools are used to check the complete sequence of the interface, but with these tools, the problem is that they can cover only executable events performed by the user. The event flow model is based on three steps. Firstly, it is based upon certain preconditions. Second, the tester shows all the sequence interactions that occur on the interface level. Thirdly, the preconditions and the effects are utilized to generate the tests. The gaps between the tools and industrial experiments are discussed in GUI model-based testing. Six gaps are highlighted by the author, scaling up the non-trivial system, reaching a sufficient coverage in a reasonable time of model extraction, validations of the extracted models, tools applicability, learning curve, and the use of tools with the existing tools, manual efforts that are utilized in minimizing the GUI testing and the maintenance effects [11].

There are various possible solutions that are also given in detail, with all the gaps identified. The outcome of event interactions depends upon the sequence. Correct GUI state leads to a useful expected screen, while incorrect states can lead to unexpected errors. Murphy tools are used for model extraction, and these model extractions are stored in a script. To the best of our knowledge, there are a lot of techniques that are used to evaluate the GUI, as the adoption of the methods and tools makes life easier for testing. Model extraction and GUI testing are the core heart of testing. Murphy tools have been used for the model extraction, and by using these tools, various GUI models have been extracted. Tools are efficient and used for commercial software applications. There is a cross-comparison between the automated tools, capture and replay, and the Murphy tool is used for model-based testing. Murphy tool reduces the test maintenance effort and is very powerful in reducing the time and effort that is used for GUI testing. The challenging aspect in using MBT(model-based Testing) is that it requires a lot of effort and complexity of models, and mapping the models to automatically generate the test cases. There is also a big difference between the GUI and dynamically executing the GUI source code to check the run-time behavior. Murphy tools have been used in organizations as well to test the complete execution of GUI [12].

The Systematic Mapping technique is used to collect all data related to GUI testing, regression testing, adequacy criteria, and coverage criteria. All the related work done in previous years in the context of GUI testing, event flow maps, and tools/techniques that are used in evaluating GUI. The main goal is to collect all the research done and gather all the data, and discuss trends that are present in these articles. To the best of our knowledge, mostly GUI testing is done on desktops, web applications and mobile platforms are missing in research. GQM is presented, which is based on some goals and questions that are related to the goals. Five goals that mainly cover the nature of articles that are available, various aspects that are available, evaluation techniques used, nature of trends, and future research directions are required to be carried out in this field.

Seven research questions are formulated to guide the study, focusing on the types of articles reviewed, their content, techniques used, test data generation methods, test oracles, developed tools or techniques, types of systems tested, evaluation methods, and whether the approaches are automated or manual. Subsequently, the selected articles, along with their citations and publication venues, are analyzed for comprehensive insights.A total of 230 articles are used in this research study, out of which 136 are used for SM. Key findings of this paper are that there are no online repositories available that contain only GUI testing-related data, so it's easy for practitioners and researchers to evaluate this process in the context of this Google Drive repository. No study has been conducted that compares the state of the art of GUI testing [13].

Test case minimization is helpful in reducing the number of redundant test cases or the test cases that become obsolete with the passage of time. Techniques such as genetic algorithms, heuristic approaches (including divide and conquer), selective redundancy, integer linear programming (DILP), cluster analysis, and set theory are used for test case minimization. The author presented a survey on the techniques that are used for test case minimization. The conclusive

part is that no technique is efficient in reducing the test case; a good technique covers both aspects which is helpful in reducing the test suite size as well as improving the fault detection deficiency [14].

Clustering is an effective approach for organizing large collections of unstructured textual data into smaller, coherent groups. This method plays a vital role in unsupervised document classification, automatic topic identification, and efficient information retrieval. Manually categorizing digital documents—such as academic articles—based on subject matter can be time-consuming and labor-intensive, making it challenging for users to sort relevant content. Clustering algorithms address this issue by automatically grouping documents according to underlying themes or topics. This becomes particularly beneficial when dealing with large datasets where processing efficiency is critical.

Clustering techniques are broadly categorized into hierarchical and partitioning methods. Hierarchical clustering builds a tree-like structure of nested clusters, while partitioning methods, such as K-means, divide data into distinct non-overlapping groups. In this study, the K-means algorithm is used to classify documents into three groups. The process involves converting documents into plain text, performing preprocessing steps (tokenization, stop-word removal, stemming, and lowercasing), and transforming them into high-dimensional vectors. Cosine similarity is then applied to measure document similarity, and the K-means algorithm clusters the documents accordingly [15].

Software testing is a critical yet challenging activity, as ineffective testing in the past has led to significant technical, social, and financial losses. It highlights that exhaustive test case generation can produce an impractically large number of redundant or infeasible test cases if the coverage criteria are not well defined. To address this issue, the study applies a hierarchical clustering approach to minimize redundancy and reduce test suite size. By grouping test cases based on requirement and branch coverage, the proposed method achieves a substantial reduction—up to 95%—by eliminating unnecessary or overlapping test cases, thus improving testing efficiency [16].

GUI testing involves several steps, including determining what to test, identifying the expected output, generating test cases, and performing verification and validation to ensure the GUI functions correctly and all user interactions are recorded. Both manual and automated approaches are used, but manual testing is resource-intensive and becomes increasingly difficult for large-scale software systems. The paper "GUI Testing: Pitfalls and Process" by Atif M. Memon highlights the growing importance of testing Graphical User Interfaces (GUIs), which now make up a significant portion of software code. It outlines key challenges in GUI testing, such as inadequate coverage criteria, lack of effective test oracles, and issues with regression testing due to frequent GUI layout changes. Traditional record-playback methods are manual and inefficient, often missing critical behaviors. The paper proposes a structured process for GUI testing that includes defining event-based coverage, generating input and expected output, step-by-step execution, and evaluating test adequacy. It emphasizes the need for automation, robust tools, and portable techniques to improve GUI testing effectiveness [17].

The pitfalls of GUI require the tools and techniques, and the results are comparable with each other. The execution of all test cases reveals the overall event coverage criteria. Regression testing is performed to check that the modified version works properly. Test designers develop a regression test suite, which consists of new test cases and the original test cases already present. Regression testing is performed to analyze the modifications in GUI objects.

## 3 Proposed methodology

We implemented a Profile Management system, which contains controls. It is a medium-sized desktop application in which all controls are extracted. The control count mechanism is implemented, in which all controls of the application are counted and saved in a test case in the directory. First, we handle all the user clicks, and these are counted. Whatever action is performed by the user is saved in a test case format. We list all controls in the form of a test case. Secondly, the event flow map represents the sequence of steps that are possible in GUI visits. Thirdly, test case clustering groups all

test cases with similar sequences, while sequence search helps locate specific test case sequences efficiently. The complete coverage report shows the coverage and how many test cases are similar to each other. Fig 1 illustrates the proposed solution, where event maps are created based on different controls. These maps are used to generate test cases, which are then analyzed for coverage and clustered using specific techniques.

We use a Profile management system, which is a medium-sized .NET desktop application that is developed to evaluate the GUI test cases and then minimize the test cases based on size and sequence. PMS handles all the personal information, sign-in information, name, address, phone number, and education info. The basic concept behind designing this application is to make a GUI where the user visits, and then the information is recorded in the form of test cases in a repository. All the GUI visits are recorded and saved in a directory, and whenever the user visits the GUI, his or her data is recorded. Let's suppose we can say that a person fills in the information his name, address, contact number, then the info is saved in the form of a text file instead of using any databases e.g SQL, MySQL. This info is useful in sequence recording and listing of all the GUI visits to give a complete GUI map.

The PMS consists of two web applications. Windows form application 1 contains the forms where the user enters his/her profile information and submits that information on clicking the submit button. Windows Form Application 2 comprises two main forms. All GUI event visits are first recorded and stored in a directory, then clustered using the K-Means data mining algorithm based on similarities among the test cases. The other strategy is document searching, where all documents are stored in a directory, and a specific sequence is searched across them using the form's search option. The matching test cases or documents are then displayed in a list view along with their sizes. Here, we can easily evaluate the long sequence containing files and the short sequence. All the files are listed in ascending and descending order. As the interface comprises various GUI components e.g text boxes, labels, buttons, radio buttons etc. These GUI components are recorded when interacted with, such as when a button is clicked or information is entered into a text box. This information is saved in a directory in the form of single test cases. The number of runs and the sequence of those test cases are saved in the directory. By using the Microsoft coded UI framework tool, the sequence is recorded and a UI map is generated, which is further used for comparing the results. Fig 2 presents the flow of the PMS case study used to evaluate the proposed approach. It illustrates multiple web applications containing user details such as sign-in, academic, and educational information. The Microsoft Coded UI Framework is employed to automate GUI testing by recording user actions, generating scripts, and replaying them to verify application behavior.

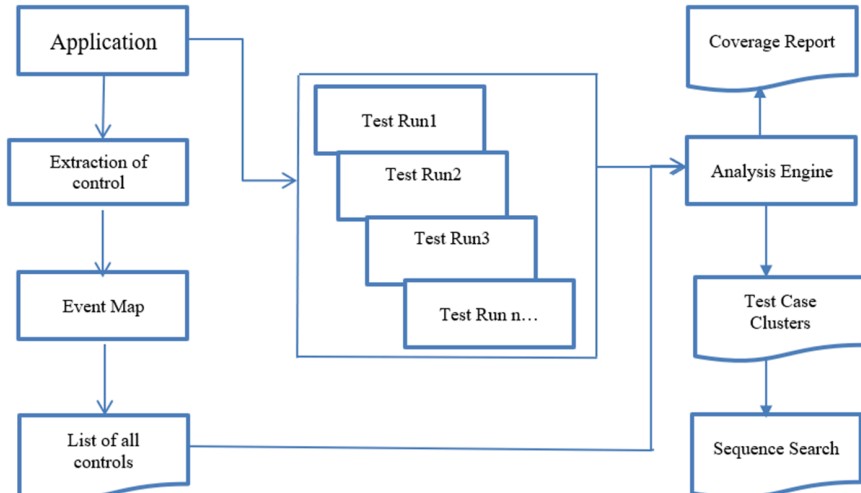

**Fig 1**. **Proposed research methodology.**

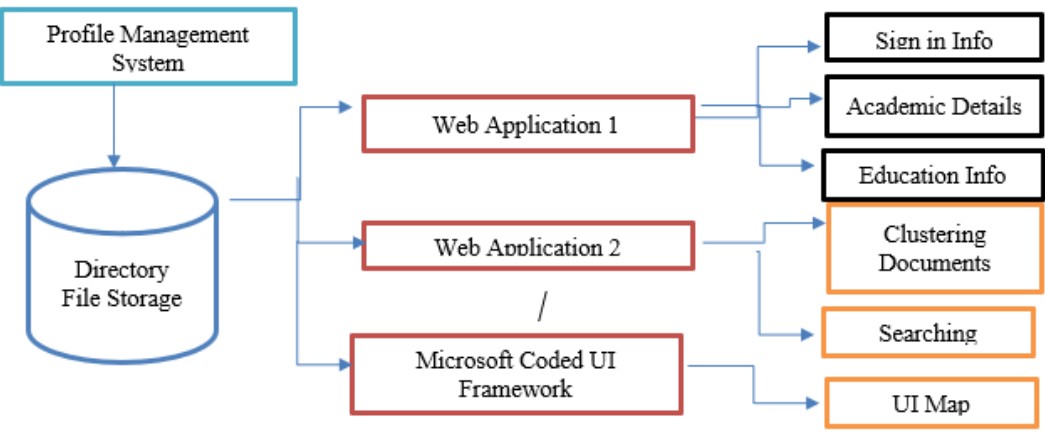

**Fig 2**. **Proposed model.**

### 3.1 Generation of event-flow graph

An event flow graph represents a set of events that are generated because of UI visits. "An event is a combination of user action and widgets". In EFG, there are nodes, and from each node, there are edges. Events are generated when the user interacts with the widget. We generated the event flow graph for each form, which helps to find the sequence of UI visits. One event can be performed more than once and all the sequences of actions are recorded in the form a test cases. There are different events e.g mouse clicks and giving input in the text box. These events can be represented by EFG. For each form, EFG depicts the visits of the user. There must be defined starting and ending points, such as button clicks. Event flow graphs for both static and dynamic GUIs are typically represented using circles to denote states, while directed arrows indicate the transitions between these states. The Profile Management System consists of three main forms: the Login Page, Personal Information, and Educational Information. Each of these forms is designed to capture and store user data. They include a variety of GUI controls such as text boxes, check boxes, radio buttons, and action buttons. The graphical interface of the application displays these three forms along with the count of controls present in each, providing a clear overview of the user input elements across the system. For each form, we generated a separate event flow graph that depicts the states and how different visits are performed. A simple login form interface includes controls such as labels, textboxes, and buttons, which are used for event handling.

Table 1 shows the Widgets and their corresponding events handled along these widgets. Textbox is used to take input from the user,labels display the information, and buttons are used for handling events.

The Event Flow Graph is shown in the Fig 3. Event Flow Graph (EFG) represents the flow of events within a graphical user interface (GUI). It shows how different user actions or events are connected and the possible transitions between them, helping in understanding and generating test cases for GUI testing.

**Table 1**. **Widgets and their corresponding events.**

| Widgets | Events |
|---|---|
| Text box | Enter Text |
| Label | — |
| Button | Click |

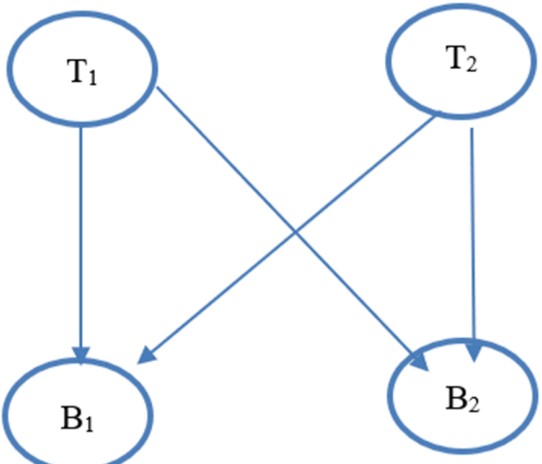

**Fig 3**. Event flow graph.

It is depicted from EFG that the user can perform various steps to visit the complete GUI. He enters information in text-box 1 and clicks Button 1, or he may enter information in text-box 2 and click Button 2, or he may take the path by entering information in Text-box 1 and clicking Button 2, or enter a password in text-box 2 and click Button 2. These events represent the sequence of GUI visits.

This is a medium-sized desktop application that is developed in the .NET Platform, and the source code is implemented in Visual Studio 2019, requiring a server. We are using a directory for saving all records instead of a database. The application can be supported by Windows Vista, Windows 7, Windows 8, or Windows 10. The processor used is Intel(R) Core(TM) i7-4700MQ CPU 2.40GHz, 2401 MHz, 4 Core(s), 8 Logical Processors, System Type is x64-based PC, supported physical memory is 8 GB.

Test cases are derived from user inputs and expected outputs to verify the correct functioning of the system. They are designed to ensure that the system behaves as intended. A test case passes when the system responds according to user requirements; otherwise, it fails. These cases help determine whether the system meets specified conditions. For this system, we developed both system-generated and manually written test cases. All test cases are documented in formal English, outlining input conditions and expected outcomes, and are presented in tabular format in the appendix.

## 4 Approach, experiments, and results

We propose a mechanism in which we record all the user interface components and organize them in a hierarchical manner. In the next step, we instrument our code in such a manner that whenever a test case is executed, we can record all the user clicks and data used as input for the execution of a test case. Since we know all the GUI components, we can find out if a component was used in a test case. A list of controls is recorded in the form of a test case, and this sequence can be compared with the CUIT by Microsoft. All sequences are comparable, and we can easily evaluate that our technique is powerful enough in test case generation and record all the sequences that are also recorded by CUIT. If we represent the set of all components as $C = \bigcup_{i=1}^{n} c = \{c_1, c_2, c_3, \ldots, c_n\}$, then each test case can be written as

$$t = \{c_1 v_1, c_2 v_2, c_3 v_3, \ldots, c_n v_n\}$$

where $c_i$ are GUI components and $v_i$ are the inputs to those GUI components, wherever applicable. We define the set $T = \biguplus_{i=1}^{n} t$ as the union of all test cases, representing the set of all GUI components that are executed at least once. All

components are said to be covered at least once if and only if $T \subseteq C \;\; \wedge \;\; C \subseteq T$, which implies $C = T$. Complete coverage means:

$$C \setminus T = \varnothing$$

We get a set containing test cases with identical subsequences. It is pertinent to note that an identical sequence can be repeated anywhere in two test cases. Consider for example, two test cases {a, b, c, d, e} and {a, b, c, d, e}, which would be reported as similar with a subsequence {a, b}. We want to find out sequences with exact similarity or a largely similar one, and check if the input was given to see if a test case is repeated with or without purpose.

We report these, and we leave it for the tester to take the necessary actions. As a side effect, we get all the sequences with similarities and select them for retesting or executing in post-evolution scenarios. This helps select test cases for regression testing.

All controls used in the application are counted using the `Count Control()` function and saved in a file named `All-Controls.txt`. Whenever a new application is executed, this file is updated to include all the controls present in the application.

The number of clicks is recorded using the `Handle. Click` method with a file stream. The file is opened, all control data is written, and the `Flush()` method is called to immediately send all output, whether in the form of HTML or C#. The results are saved using this method.

Finally, the `Close()` method is used to properly close the file. The `System.IO` namespace is used to handle file reading and writing operations. The `FileInfo` class provides detailed information about files, while the `FileStream` class gives access to stream-based file operations.

Next, we introduce two methods that allow us to record which controls are clicked and what inputs are provided during test case execution. We maintain a separate file containing all available controls, while another file stores the controls and inputs used in each test case. To achieve this, we set up and handle click events that automatically record the control interactions in a structured test case format. This approach allows us to record which controls are clicked and the values passed to them during subsequent interactions. In a nutshell, we implement a listener that records events whenever a control unit (e.g., a button, text box, or menu item) is clicked. Additionally, we capture input data whenever a control takes user input. This allows us to understand the structure and format of test cases more clearly. We store control information and the input provided to those controls separately. This separation is intentional and designed to improve the execution efficiency of our algorithm. Once we identify two sequences that are the same, we only need to analyze the inputs provided to these sequences in order to conduct a detailed comparison. We record the components used and the inputs applied during test case execution. This data is valuable for test case minimization. Next, we apply an algorithm to detect similarities within test sequences, which is detailed in the following section of Algorithm 1. This algorithm systematically finds and groups test case sequences that share common patterns of increasing lengths, which helps in identifying frequently occurring event or action sequences in GUI testing.

The file `run_num.txt` contains all records of actions performed within the application. Each time the file is opened, the number of runs is tracked and saved. The variable `filenum` is incremented by 1 with each subsequent run, helping distinguish the logs of different executions. Algorithm 2 shows how various GUI controls are used in test case generation techniques.This algorithm automatically tracks user clicks on GUI controls, records their names, and stores them as part of test cases. It forms the basis for automated GUI test case generation.

## 4.1 Setup clicks and event handling

Set up clicks and handle clicks are used to record events from the Graphical User Interface (GUI). Event handlers are utilized within the application to manage these events. In software development,events are defined as interrupts generated by specific user actions, such as clicking a button or entering text into a text box.

**Algorithm 1 Sequential pattern mining.**

**Input:** `Set` $T$ `of test case sequences`

**Output:** `Set` $Ts$ `of sequences with` $n$ `common subsequences`

1. `Initialize` $n \leftarrow 2$
2. `For each sequence` $s_i$ `in` $T$:
  (a) `For each sequence` $s_j$ `in` $T$:
   i. `If both` $s_i$ `and` $s_j$ `contain a common subsequence` $s_k$ `such that length of` $s_k = n$, `then:`
     A. `Add` $s_i$ `and` $s_j$ `to` $Ts$
  (b) `Increment` $n$ `by 1`

**Algorithm 2 Capturing GUI control interactions for test case generation.**

**Input:** `Set of controls`

**Output:** `Set of test cases containing controls and control clicks`

1. `For each setup click event:`
  (a) `Attach event handler: Control.Click += Handle. Click`
2. `For each handle click:`
  (a) `Assign control name: Control = Control.Name`
3. `For counting controls:`
  (a) `Output the control name: Controls.WriteLine(Control.Name);`

All these events are:

- Counted
- Controlled
- Handled

This comprehensive recording mechanism ensures accurate test case generation and allows for refined sequence comparison in our algorithm.

### 4.2 Clustering

Clustering is a technique used to group similar items into a single category. It is a process of organizing similar members into groups and is considered an **unsupervised learning technique**. Various data mining algorithms are applied to identify and group related items based on similarity.One common approach is **flat clustering**, also known as *exclusive clustering*, where similar data items are grouped without overlapping. In the context of this project, all repeating items that contain the same sequence sets are grouped together for efficient analysis. All the files to be clustered are stored in text format within a specific directory. A `FolderBrowserDialog` is used to navigate and select the directory, and the retrieved files are displayed in a `List Box`. Upon a button click, these files are successfully added and made ready for clustering. When the clustering process starts, the selected files are clustered based on their sequence content. The **K-Means algorithm** is used to cluster the files. This algorithm is efficient for large datasets and works by calculating the median and mode of item sequences. A custom `Document Clustering` class is used to implement the core logic of

clustering. Clustering is useful in combining similar group items based on sequence similarity, and the basic steps of the K-Means algorithm are as follows:

1. Select $K$ number of clusters (where $K$ is a constant).
2. Choose $K$ initial objects based on the starting sequence.
3. Repeat the following steps:
   (a) Assign each object to its nearest cluster.
   (b) Recalculate new clusters.
   (c) Compute the mean points of each cluster.
4. Continue repeating the above steps until there is no change in the cluster assignments.

Fig 4 shows K-Means Technique is applied and documents are clustered into groups. K-Means organizes documents into clusters so that documents within a cluster are more similar to each other than to those in other clusters.

   An algorithm partitions the object based on its attributes/features, and similarity is calculated using methods such as cosine similarity and Euclidean distance. Fig 5 shows K-Means Technique is applied and documents are clustered in a group.

   All listed files are available in the directory are searched and listed in a `List Box` with their names. The files are added up and are clustered on button click. The documents are clustered based on the sequence present. The same

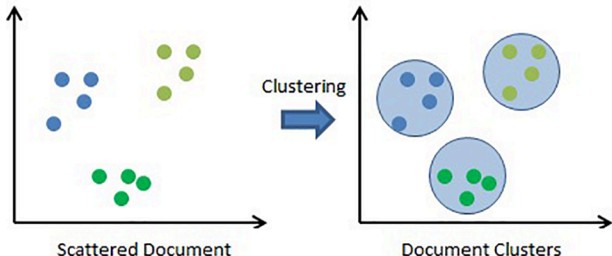

**Fig 4**. **K-Means document clustering.**

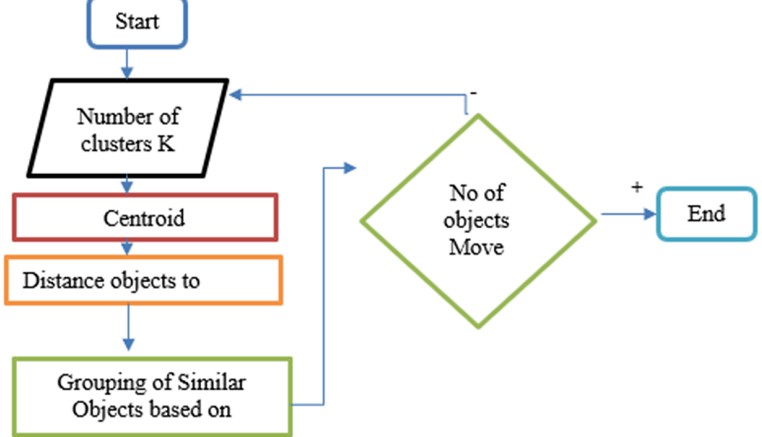

**Fig 5**. **K-Means document clustering flow chart.**

sequence of documents is grouped in one cluster. The term frequency shows how many times a particular sequence occurs in a document, and the sequence is similar or different. The similarity score is used to calculate the similarities between the sequences. As shown in Algorithm 3, documents are clustered using cosine similarity.The algorithm groups documents into clusters based on cosine similarity, meaning documents that share similar content or patterns are grouped together. It iteratively updates cluster centers (like in K-Means) until stable, well-defined clusters of related documents are formed.

**Algorithm 3 Clustering of documents using cosine similarity.**

**Input:** Document set to be clustered based on sequence similarity **Output:** Documents grouped into clusters

```
1. For each centroid computation:
  (a) Group documents based on similarity
  (b) Add the centroid to the centroid collection
2. Repeat until the stopping criteria are met:
  (a) For each document vector in the document collection:
     i. Find the closest cluster using similarity (e.g., cosine similarity)
    ii. Add the document to the corresponding grouped cluster
  (b) Calculate new centroid (mean points) for each cluster
  (c) Check stopping criteria:
    • If the stopping criteria are not met, initialize the result set for the next
      iteration
```
3. For each cluster (indexed by $i <$ cluster.count):
```
  (a) Compute cosine similarity for documents
  (b) Group similar documents accordingly
4. For each similarity measure:
  (a) If similar documents are found:
    • Assign the document to the cluster with the lowest index
    • Return the cluster index
5. For each document vector space in a grouped cluster:
6. Compute the new cluster center:
```

$$\text{ClusterCenter.GroupedDoc.VectorSpace} = \frac{\text{Total}}{\text{ClusterCenter.GroupedDoc.Count}}$$

### 4.3 Search sequence in a document

A search operation is performed on all the listed documents, during which the sequence file is scanned, and the matching results are displayed in the list box. It can read text files placed in a directory by using the specified functions through the delegate Match. This function is used for iterations and looking for documents in a list. It searches for the document case-sensitively and the matching document where the sequence is matched. We have two classes text reader and a text-to-string converter class, which read the text file and convert the content to strings. First, we read the document from the directory by using the Read Document class, and the String builder class is used to concatenate the content in the text document. Then the new line is appended by using the append line. Content is appended to the document element.

All files are searched from the directory and the subdirectory and added to the List box control, where the sequence of files is matched. The files where the search string is matched and then these files are returned in a list. Two operations are performed, either a string or a regex. All the listed files from the subdirectories are listed with their file size.

We demonstrate the effectiveness of our approach through this example, where we can evaluate that there is a requirement to have new test cases for comprehensive coverage. We also know which of the controls need to be involved to complete the coverage. However, our approach has certain limitations. We do not achieve complete coverage, and even with additional test cases, it is not possible to include every possible combination of controls within the application. It is also pertinent to note that our algorithm that reports similarities is neither efficient nor it is conclusive in any manner. We, however, find out sequence clusters and the sequences with total similarities of our interest. This is important for us in the event of regression testing, as we can find out which of the test cases involve a control. This will help us to find the impact of change in the implementation of controls, e.g buttons, and we can find out the impact of evaluation. Fig 6 shows the document searching flow chart in which the user enters the sequence and respective valid/invalid test cases are generated.

This Algorithm 4 searches all files in a given directory and its subdirectories for a specific sequence. If a match is found, the file is opened and listed in the GUI; otherwise, a "no match" message is returned.

CUIT is given by Microsoft, and it can be easily integrated with Team Foundation Server. This is a record and playback tool and is helpful in the generation of UI map, which contains all the sequences and subsequent values that are passed to the controls to check functionality. It is a powerful tool and gives a complete UI map using generated code. CUIT supports both web and Windows applications. There are three main parts of the GUI controls that are generated by using coded UI. UIMap.designer.cs, UIMap.cs and UIMap.uitest. The first two files are physical files for partial classes, and the third one is an XML equivalent that records all the actions with the help of the CUIT builder. Coded UI test is the automated way of testing interfaces. It can run all the visits of user. The tool is added in the running project and executed. UI visits are recorded in the UI Map. uitest files. The number of controls and what values are passed to these controls are saved in UIMap.uitest files. Coded UI is used for the action recordings of the user. Through the Code UI test builder, the sequence of actions is recorded. A Test Method is created that contains all the sequence of actions by the user. The user interface of the login page is used to capture the sequence of user actions through coded UI testing. These actions are saved as a recorded method, where user interactions are automatically captured and translated into executable code that represents the complete sequence of operations. The recording mechanism tracks all user activities during interaction with the interface and generates corresponding code. The UI map stores the full sequence of user actions, enabling automated playback and validation of the workflow.

Sequence represents any actions that are performed e.g when the user clicks a button, enters text, selects a checkbox, etc. All events represent a type of sequence, which is the path that is taken by the user. To the best of our knowledge, we use an approach to list all the possible events. We execute all the test cases that are in the test suite to test the application. We get a set of files generated as a side effect of the test case in terms of the controls used and what the inputs

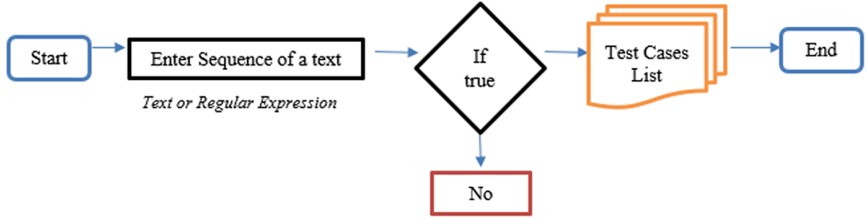

**Fig 6**. Document searching flow chart.

**Algorithm 4 File search algorithm.**

**Require:** Directory path, search sequence
**Ensure:** List of files containing the search sequence
 **Input:** A directory with files and subdirectories, and a search sequence
 **Output:** List of matched files displayed in GUI

1: Initialize an empty list matchedFiles

2: Retrieve all file paths from the directory and its subdirectories

3: **for** each filePath in the list of retrieved files **do**
4: **if** File.Exists(filePath) **then**
5: **if** textDoc.IsMatch(filePath) **then** ▷ Check for sequence match
6: Add filePath to matchedFiles
7: Open the file using Process.Start(filePath)
8: Add the file to GUI ListView
9: **end if**
10: **else**
11: **return** "File not found"
12: **end if**
13: **end for**

14: **if** matchedFiles is not empty **then**
15: **return** matchedFiles
16: **else**
17: **return** "No matching files found"
18: **end if**

are for the execution. After all the analysis, we reported the coverage of each test case. It is worth noting here that we are not able to achieve coverage even after the execution of each test case. It is important to note that coverage even after executing longer sequence test cases, is not achievable, and our system has reported many similarities. We have forms that contain several controls. The sequence of events is generated when multiple users visit the interface, and a repeated sequence may occur, which results in a similar type of event. Repeated sequence sets can be eliminated at the time of test case minimization. In the results section, we presented three forms, and their coverage report shows control occurrences and similarities that exist between the sequence of events. All possible sequences are listed that the user performs, and all these sets of events are recorded and shown in the form of event flow graphs. The user can perform a set of visits; these visits can be performed in sequence or repeated sequence generation form. As a user can do multiple visits also so the sequence can be repeated and listed. These redundant visits are repeated test cases that will be removed during test case minimization. All the possible interactions of the user are listed in the form of a set collection for subsequent forms. Our result section is organized in a way that firstly, we mentioned the maximum possible GUI visits that are recorded in event generation. These visits are mapped clearly in EFG, which shows the sequence of visitors. Then these sequence and subsequences is shown in the tabular form, which contains sets and subsets. All the subsequences are part of the sequence and can be eliminated during test case minimization because this results in repeated sequence generation and similarities among the test cases. There are many redundant test cases that are present, and these can be easily removed during testing. Then the coverage report shows the probability of occurrence of the controls. This approach of test case generation and sequence recording can be easily compared with Microsoft's test case generation. The map also gives the same sequence of actions with its recorded methods.

The system consists of three forms, and all controls present on these forms are listed in the subsequent tables. All possible sequences and subsequences of user interactions are recorded in a separate table, representing the sets and subsets corresponding to the maximum possible visits to the user interface. The subsets are derived from the larger sets, and each subset represents a sequence that is part of a complete interaction set. These repeated sequences can be effectively removed during test case minimization, leaving only unique interaction sequences for further analysis and execution. The coded UI–based login interface utilizes multiple controls, and user events are recorded to capture interaction behavior for automated testing.

### 4.4 Controls and sequence coverage criteria

We consider Fig 3 with the following controls: T1, T2, B1, B2. The test sequences executed are:

- {T1, T2, B1}
- {T1, T2, B2}
- {T1, B1}
- {T1, B2}
- {T2, B1}
- {T2, B2}

We execute all test cases in the test suite for application testing. This generates a set of files as a side effect of test execution, which precisely document:

- The format of each test case in terms of controls used
- The specific inputs used for execution

### 4.5 Coverage report and similarities

We find out and report coverage after the execution of each test case in the coverage report table. We could not achieve full coverage despite executing multiple sequence test cases, as the system reported many similarities.

### 4.6 Value for regression testing

Despite these limitations, we find sequence clusters and sequences with total similarities that are of particular interest. This proves valuable for regression testing as it enables us to:

- Identify which test cases involve specific controls
- Assess the impact of changes to control implementations (e.g., buttons)
- Evaluate the impact of system evolution

Table 2 summarizes the test case sequences along with their occurrence frequency, coverage, and similarity values. Test cases **T3–T6** show higher coverage (50%), indicating repeated or broader interaction sequences, while **T1–T2** have lower coverage (25%), representing unique or less frequent interactions. The similarity column highlights the relationship between test cases based on their event sequence patterns.

Sequence $\{\{1, 2, 1\}, \{1, 1\}\}$, $\{\{1, 2, 2\}, \{1, 2\}\}$, $\{\{1, 2, 1\}, \{2, 1\}\}$, $\{\{1, 2, 2\}, \{2, 2\}\}$ occur repeatedly and there are similarities exist so these can be removed during test case minimization and unique sequence of control is left behind. This creates ease for the tester to find unique sequence test cases. As the larger sets contain subsets with repeated occurrences of controls. The same group of sequence events is clustered using the K-means clustering algorithm.

**Table 2**. **Test case sequences and their similarity analysis.**

| S# | Test Cases | Sequences | No. of Occurrences | Coverage | Similarity |
|---|---|---|---|---|---|
| T1 | 1–10 | {T1T2B1} | 1 | 25% | {1,2,1} |
| T2 | 1–20 | {T1T2B2} | 1 | 25% | {1,2,2} |
| T3 | 1–30 | {T1B1} | 2 | 50% | {{1,2,1}, {1,1}} |
| T4 | 1–40 | {T1B2} | 2 | 50% | {{1,2,2}, {1,2}} |
| T5 | 1–50 | {T2B1} | 2 | 50% | {{1,2,1}, {2,1}} |
| T6 | 1–60 | {T2B2} | 2 | 50% | {{1,2,2}, {2,2}} |

### 4.7 Clustering of text files

If we have six documents that contain the sequences {ABC}, {code}, {egg}, {haj}, {abc}, {abcd}, then the sequences containing `abc` are clustered into one group. Based on the sequence, the number of iterations is computed, and the test cases are clustered into their respective groups.

For example, suppose there is a sequence of test cases: {`T1T2B1, T1T2B2, T1, T2, B1, B2, T1T2, T1B1, T2B1, T2B2, T1T2B1, T1T2B2`}. This sequence contains repeated patterns, and these repeated sequences can be clustered into one group depending on the user visits.

Clustering is useful for grouping similar test cases that share the same execution sequence. K-means is a fast and simple algorithm that is well-suited for grouping large datasets. In this approach, text documents representing test cases are clustered based upon similarity, where the clustering process considers the number of documents involved and iteratively groups them until convergence is achieved.

### 4.8 Sequence search in a text document

The search sequence in a text document identifies how often a specific control, such as a text box, appears across different test cases. This process determines the number of test cases in which the control occurs and presents the result through the generated user interface output.

### 4.9 Sequences generated through Microsoft coded UI framework

This framework can also be used to record interactions with interface components. The UI Map and UI test files are stored in XML format and contain all possible interactions between the user and the interface components. In the considered web application, three forms are used, and the maximum possible user interface interactions are captured through the coded UI framework. All recorded methods illustrate how frequently each control is used during sequence recording, after which a complete user interface map is generated to represent these interactions.

## 5 Conclusions and future work

The research introduces a control count mechanism that records GUI visits as test cases. A .NET desktop application was developed to manage two web applications — one for recording GUI sequences and another for clustering similar sequences and grouping them for analysis. It also includes a sequence/document search tool to help testers easily locate specific test cases among thousands.

This approach simplifies tracking user interactions, enables efficient coverage analysis, and supports test case minimization by eliminating duplicates or redundant cases. It also enhances regression testing by helping testers track changes when components are modified.

The results show that the proposed mechanism performs comparably to Microsoft's Coded UI framework, accurately capturing all events. It addresses common GUI testing challenges such as scalability, controllability, portability, and modifiability.

The main contribution lies in minimizing test cases at both unit and system levels, reducing overlap and improving testing efficiency. Future work aims to extend this technique by testing multiple applications, enhancing algorithms, and expanding to web and Android platforms to achieve broader coverage and deeper analysis of GUI events.

## Author contributions

**Conceptualization:** Raheela Ambreen, Tamim Ahmed Khan.

**Data curation:** Raheela Ambreen.

**Formal analysis:** Raheela Ambreen, Tamim Ahmed Khan.

**Funding acquisition:** Raheela Ambreen.

**Investigation:** Raheela Ambreen, Tamim Ahmed Khan.

**Methodology:** Raheela Ambreen, Tamim Ahmed Khan.

**Project administration:** Raheela Ambreen.

**Resources:** Raheela Ambreen.

**Software:** Raheela Ambreen.

**Supervision:** Tamim Ahmed Khan.

**Validation:** Raheela Ambreen.

**Visualization:** Raheela Ambreen, Tamim Ahmed Khan.

**Writing – original draft:** Raheela Ambreen, Tamim Ahmed Khan.

**Writing – review & editing:** Raheela Ambreen.

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
