## [Decision Letter · Decision Letter 0]

8 Oct 2025

PONE-D-25-45331GUI TEST CASE MINIMIZATION USING SEQUENCE MININGPLOS ONE

Dear Dr. Khan,

Thank you for submitting your manuscript to PLOS ONE. After careful consideration, we feel that it has merit but does not fully meet PLOS ONE’s publication criteria as it currently stands. Therefore, we invite you to submit a revised version of the manuscript that addresses the points raised during the review process.

We look forward to receiving your revised manuscript.

Kind regards,

Sajid Anwar, Ph.D

Academic Editor

PLOS ONE

3. We note that your Data Availability Statement is currently as follows: [All relevant data are within the manuscript and its Supporting Information files]

Additional Editor Comments (if provided):

Reviewers' comments:

Reviewer's Responses to Questions

**Comments to the Author**

1. Is the manuscript technically sound, and do the data support the conclusions?

Reviewer #1: Yes

Reviewer #2: Yes

2. Has the statistical analysis been performed appropriately and rigorously?

Reviewer #1: Yes

Reviewer #2: Yes

3. Have the authors made all data underlying the findings in their manuscript fully available?

Reviewer #1: Yes

Reviewer #2: Yes

4. Is the manuscript presented in an intelligible fashion and written in standard English?

Reviewer #1: Yes

Reviewer #2: Yes

5. Review Comments to the Author

Reviewer #1: The paper needs the minor revision. Please go through the comments to modify the paper accordingly. The manuscript is accepted with minor revisions. There are some grammatical and sentence structure mistakes. Besides, the methodology section should also be modified, and there have to be the in-text citations in the manuscript. Specifically, the Introduction section lacks any such support. Also the updated references are missing in the manuscript. For details, please see the comments file attached.

Reviewer #2: This work proposes a sequence recording technique combined with K-Means clustering to minimize GUI test cases while maintaining coverage. A medium-sized desktop application was used as a case study, where event flows, user interactions, and control usage were recorded, clustered, and analyzed for redundancy. The results show that the approach effectively reduces redundant test cases, supports regression testing, and can be extended to broader GUI testing contexts. Overall, the current work significantly improves the body of knowledge in the regression testing domain. The following are some section-wise comments to further enhance the quality of the work:

• Abstract:

•It is useful to streamline the flow. Currently, the abstract is slightly long and repeats "representative set" twice. Consider condensing sentences for a sharper impact.

•Example: “We propose a sequence recording technique that reports coverage and applies K-Means clustering to minimize test cases, achieving comparable requirement coverage with fewer cases.”

• Introduction:

•Some sentences are lengthy and could be broken into two for readability (e.g., lines 8–11 about test case generation being time-consuming).

•Ensure consistent terminology: sometimes you use GUI visits, GUI events, and GUI interactions. Standardize.

• Related Work (Section 2):

•References [1]–[14] are summarized well but could benefit from shorter, more concise sentences. At times, the summaries are verbose (e.g., lines 83–96).

•Check citation consistency: IEEE style usually uses et al. after the first author's name. Ensure formatting aligns with PLOS ONE style.

• Methodology (Section 3):

•Figures: Add brief descriptive captions to make them self-explanatory (e.g., Fig. 1 “Overall Research Methodology for GUI Test Case Minimization”).

•Clarify acronyms on first use (e.g., CUIT, PMS).

• Results (Section 4):

•Tables and figures: Ensure uniform formatting. Table 2 spacing and labels look slightly inconsistent.

•Highlight key findings more explicitly in text instead of leaving them implied in the table (e.g., “Sequences T3–T6 show 50% coverage and can be minimized by eliminating redundant subsets”).

• Conclusion (Section 5):

•Add a stronger forward-looking statement (e.g., about applying the approach to web/mobile apps, which you already mentioned briefly).

•Correct capitalization in “future work” → “Future work.”

• Language / Style:

•Watch for minor grammar issues, e.g.,

o“this application is fair enough for tracking” → “this application is suitable for tracking.”

o“like some of the events” → “for example, some of the events.”

•Avoid informal words like fair enough, like, etc. in academic writing.

• Formatting:

•Check uniform font and spacing in algorithms (Algorithm 1–4). Some indentation is inconsistent.

•Ensure all acronyms are defined and consistently used.

6. PLOS authors have the option to publish the peer review history of their article (what does this mean?). If published, this will include your full peer review and any attached files.

Reviewer #1: No

Reviewer #2: No

---

## [Author Response · Author response to Decision Letter 1]

27 Nov 2025

Title: GUI TEST CASE MINIMIZATION USING SEQUENCE MINING

Dr Tamim Ahmed Khan & Engr.Raheela Ambrin

Reviewers Comments Authors Answer Page Number/Section number

1- Reviewer #1: The paper needs the minor revision. Please go through the comments to modify the paper accordingly. The manuscript is accepted with minor revisions. There are some grammatical and sentence structure mistakes. Besides, the methodology section should also be modified, and there have to be the in-text citations in the manuscript. Specifically, the Introduction section lacks any such support. Also the updated references are missing in the manuscript. For details, please see the comments file attached.

The entire manuscript is proofread, and all pages are revised for English language and style following reviewers’ comments. in all pages

2- Reviewer #2: This work proposes a sequence recording technique combined with K-Means clustering to minimize GUI test cases while maintaining coverage. A medium-sized desktop application was used as a case study, where event flows, user interactions, and control usage were recorded, clustered, and analyzed for redundancy. The results show that the approach effectively reduces redundant test cases, supports regression testing, and can be extended to broader GUI testing contexts. Overall, the current work significantly improves the body of knowledge in the regression testing domain. The following are some section-wise comments to further enhance the quality of the work:

Abstract:

•It is useful to streamline the flow. Currently, the abstract is slightly long and repeats "representative set" twice. Consider condensing sentences for a sharper impact.

•Example: “We propose a sequence recording technique that reports coverage and applies K-Means clustering to minimize test cases, achieving comparable requirement coverage with fewer cases.” The manuscript is updated. The abstract is summarized, and some experimental stats are given which shows the overall percentages.

Page 1

3. Some sentences are lengthy and could be broken into two for readability (e.g., lines 8–11 about test case generation being time-consuming).

Ensure consistent terminology: sometimes you use GUI visits, GUI events, and GUI interactions. Standardize The presentation of the manuscript is improved, and lengthy sentences are broken down.

Consistent terminologies are mentioned in the revised manuscript. Lines 8-11

4- References [1]–[14] are summarized well but could benefit from shorter, more concise sentences. At times, the summaries are verbose (e.g., lines 83–96) References are summarized and some recent research papers are added In the related work section Line 103-122,272-280,281-289

5- Methodology (Section 3):

Figures: Add brief descriptive captions to make them self-explanatory (e.g., Fig. 1 “Overall Research Methodology for GUI Test Case Minimization”). All figures and tables mentioned in the paper are elaborated and completely explain the idea.

Figure 1 -326,361,383,385,391,521,525,567,633,664,683,688,698.

Tables and figures: Ensure uniform formatting. Table 2 spacing and labels look slightly inconsistent.

•Highlight key findings more explicitly in text instead of leaving them implied in the table (e.g., “Sequences T3–T6 show 50% coverage and can be minimized by eliminating redundant subsets”)

Table 2 spacing is justified, and the elaborated and highlighted section shows the coverage in percentages. Findings are highlighted Page 664

Add a stronger forward-looking statement (e.g., about applying the approach to web/mobile apps, which you already mentioned briefly).

•Correct capitalization in “future work” → “Future work.” Future work is summarized and concise and clear Line 699-718

• Language / Style:

•Watch for minor grammar issues, e.g.,

o“this application is fair enough for tracking” → “this application is suitable for tracking.”

o“like some of the events” → “for example, some of the events.”

•Avoid informal words like fair enough, like, etc. in academic writing.

• Formatting:

•Check uniform font and spacing in algorithms (Algorithm 1–4). Some indentation is inconsistent.

•Ensure all acronyms are defined and consistently used. Sentences are rephrased, and overly long sentences are broken down. Informal words are avoided, and font and spacing are justified.

Keywords are also highlighted in the paper. Line 1-4

---

## [Decision Letter · Decision Letter 1]

15 Dec 2025

GUI TEST CASE MINIMIZATION USING SEQUENCE MINING

PONE-D-25-45331R1

Dear Dr. Khan,

We’re pleased to inform you that your manuscript has been judged scientifically suitable for publication and will be formally accepted for publication once it meets all outstanding technical requirements.

Kind regards,

Sajid Anwar, Ph.D

Academic Editor

PLOS One

Additional Editor Comments (optional):

Reviewers' comments:

Reviewer's Responses to Questions

**Comments to the Author**

1. If the authors have adequately addressed your comments raised in a previous round of review and you feel that this manuscript is now acceptable for publication, you may indicate that here to bypass the “Comments to the Author” section, enter your conflict of interest statement in the “Confidential to Editor” section, and submit your "Accept" recommendation.

Reviewer #1: All comments have been addressed

2. Is the manuscript technically sound, and do the data support the conclusions?

Reviewer #1: Yes

3. Has the statistical analysis been performed appropriately and rigorously?

Reviewer #1: N/A

4. Have the authors made all data underlying the findings in their manuscript fully available?

Reviewer #1: Yes

5. Is the manuscript presented in an intelligible fashion and written in standard English?

Reviewer #1: Yes

6. Review Comments to the Author

Reviewer #1: (No Response)

7. PLOS authors have the option to publish the peer review history of their article (what does this mean?). If published, this will include your full peer review and any attached files.

Reviewer #1: No

---

## [Editor Report · Acceptance letter]

PONE-D-25-45331R1

PLOS One

Dear Dr. Khan,

I'm pleased to inform you that your manuscript has been deemed suitable for publication in PLOS One. Congratulations! Your manuscript is now being handed over to our production team.

Kind regards,

on behalf of

Dr. Sajid Anwar

Academic Editor

PLOS One